# Lossless enrichment of trace analytes in levitating droplets for multiphase and multiplex detection

Xueyan Chen[1,2,6], Qianqian Ding[1,6], Chao Bi[3], Jian Ruan [2] ✉ & Shikuan Yang [1,2,4,5] ✉

Concentrating a trace amount of molecules from liquids, solid objects, or the gas phase and delivering them to a localized area are crucial for almost any trace analyte detection device. Analytes within a liquid droplet resting on micro/nanostructured surfaces with liquid-repellent coatings can be concentrated during solvent evaporation. However, these coatings suffer from complex manufacturing procedures, poor versatility, and limited analyte enrichment efficiency. Here, we report on the use of an acoustic levitation platform to losslessly concentrate the analyte molecules dissolved in any volatile liquid, attached to solid objects, or spread in air. Gold nanoparticles can be simultaneously concentrated with the analytes in different phases, realizing sensitive, surface-enhanced Raman scattering detection even at attomolar ($10^{-18}$ mol/L) concentration levels. The acoustic levitation platform-enabled, lossless analyte enrichment can significantly increase the analytical performance of many conventional microsensing techniques.

Ultrasensitive detection of trace analytes is important in a broad range of fields, ranging from analytical chemistry[1] to disease diagnostics[2–5], biomedicine[6–9], environmental science[10–12], and national security[13,14]. In these practical application fields, a trace amount of multiple analytes is either dispersed in water or organic liquids, attached to solid surfaces, or dispersed within a gaseous mixture[3–7,12–15]. Surface-enhanced Raman scattering (SERS) is promising in trace analyte detection because of its high sensitivity, label-free detection, and miniaturization[16–25]. Various SERS substrates integrated with dense and uniformly distributed, ultrasensitive SERS sites (known as "hot spots" and usually located at <10 nm gaps between adjacent nanoparticles) have already been fabricated[24–28]. The working area of the SERS sensors, which is generally at the level of a few square micrometers, is determined by the spot size of the excitation laser beam[12]. To address the enriched analyte aggregate under a microscope during SERS measurements, concentrating and driving the analytes in different phases into hot spots

ideally located at the level of hundreds of square micrometers is desirable but has been proven challenging[12,26,29,30]. The pinning of the solid/liquid/gas three-phase contact line caused the accumulation of analyte molecules around the pinning circle, which is known as the "coffee ring" effect[31]. Contact line pinning may be greatly delayed on the liquid repellent surfaces (e.g., superhydrophobic and super-oleophobic surfaces, molecularly smooth slippery surfaces) but is inevitable[32]. The slippery liquid-infused porous surfaces (SLIPSs) can enrich analytes from common liquids after solvent evaporation, avoiding the "coffee ring" effect, whereas the lubricant tends to wrap the liquid droplet and influence the SERS detection[12] (Supplementary Fig. 1). Actually, it is impossible to enrich all of the analytes into the final aggregate on any slippery surface, induced by either the wetting defects and/or the evaporation process (Supplementary Fig. 2). Additionally, all of the existing surfaces are only applicable to nonsticky solutions and have limited analyte enrichment efficiency for sticky

[1]Institute for Composites Science Innovation, School of Materials Science and Engineering, Zhejiang University, Hangzhou 310027, China. [2]Department of Medical Oncology, The First Affiliated Hospital, Zhejiang University School of Medicine, Hangzhou 310003, China. [3]Core Facilities, Zhejiang University School of Medicine, Hangzhou 310003, China. [4]State Key Laboratory of Fluid Power and Mechatronic Systems, Zhejiang University, Hangzhou 310027, China. [5]Baotou Research Institute of Rare Earths, Baotou 014030, China. [6]These authors contributed equally: Xueyan Chen, Qianqian Ding. ✉ e-mail: software233@zju.edu.cn; shkyang@zju.edu.cn

solutions (Supplementary Fig. 3). Optical trapping and other concentration methods have been developed to concentrate analytes/nanoparticles in bulk solutions[33–35], whereas it is difficult to concentrate all of the analytes from the solutions. Methods capable of losslessly enriching analytes in any volatile liquid and from tiny solid objects are still unavailable.

Acoustic levitation, namely, levitating objects in various media (e.g., air) using acoustic radiation forces[36–39], has been investigated in metallic solidification[40], contactless transport of matter[41,42], drop dynamics[43] and analytical chemistry[44–47] for many decades. The SERS technique was selected to monitor the chemical reaction and crystallization process within a levitating drop in 2003 by Prof. Lendl's group and Prof. Nilsson's group, respectively[48,49]. Here, we report lossless analyte enrichment in any phase using a droplet levitating enrichment (DLE) platform. Au nanoparticles as SERS enhancers are introduced to a levitating analyte solution droplet, simultaneously enriching with the analytes to realize ultrasensitive SERS detection. The limit of detection (LOD) can be pushed down simply by increasing the starting volume of the levitating droplet. The DLE platform can be combined with almost any sensing technique collecting signals from microscale areas (or microsensing) methods (e.g., photoluminescence, near-infrared absorption), showing great application potential in trace analyte detection fields.

## Results

### Analyte enrichment capability of the DLE platform

The sound pressure of an acoustic wave generated by a piezo-electronic transducer (frequency: 20.7 kHz) can compensate for the gravitational force of the droplet and thus realize droplet levitation[50,51] (Fig. 1a). The challenging lossless, multiplex, and multiphase analyte enrichment for slippery surfaces became simple and straightforward for the DLE platform. Multiple analytes in the solvents can be simultaneously concentrated with the Au nanoparticles during solvent evaporation for consequent SERS detection. Aqueous solutions were introduced to the organic phase (e.g., toluene) by a pipette tip, achieving multiphase analyte enrichment (Supplementary Movie 1 and Fig. 1a). Experimentally, the droplet was squeezed into an ellipsoid shape to provide enough force to balance the gravitational force[52] (Fig. 1a). As evaporation proceeded, the ellipsoid droplet gradually became quasi-spherical. At this moment, we needed to slightly reduce the distance between the acoustic emitter and the reflector to squeeze the quasi-spherical droplet back into the ellipsoid shape. When the droplet was very small (e.g., 2 nL in volume), it oscillated at a high frequency. We should transfer the tiny droplet onto a substrate for further characterization. The successful transfer rate for a 10 nL droplet was 100%, while for a 2 nL droplet, it was ~70% for our apparatus. A 10 µL transparent droplet composed of 10 µM crystal violet (CV) dye molecules was enriched into an ~100 µm violet droplet (~0.5 nL in volume) after solvent evaporation, easily achieving a concentration increase of 20000 times (Supplementary Movie 2 and Fig. 1b). When the ratio of the hydrostatic pressure to the capillary pressure exceeds ~1.5, the droplet will disintegrate and explode[44]. Therefore, organic droplets with low surface tensions tend to explode. The largest droplet of water/ethanol that can be levitated for our equipment can reach 180 µL/30 µL, corresponding to a concentration increase of 360,000/60,000 times. Increasing the power and adjusting the acoustic frequency of the acoustic levitator can expand the size range of the droplet that can be levitated[37], further increasing the analyte enrichment capability.

In addition to the analyte molecules, microparticles with different shapes, nanoparticles, and ions can also be concentrated using an acoustic levitator (Fig. 1c). The 500 nm polystyrene spheres dispersed in ethanol assembled into a large ball after ethanol evaporated (Supplementary Movie 3). The outermost layer of the ball comprised hexagonally arranged polystyrene spheres. In contrast, the 300 nm-sized metal-organic-framework (MOF) octahedra and truncated

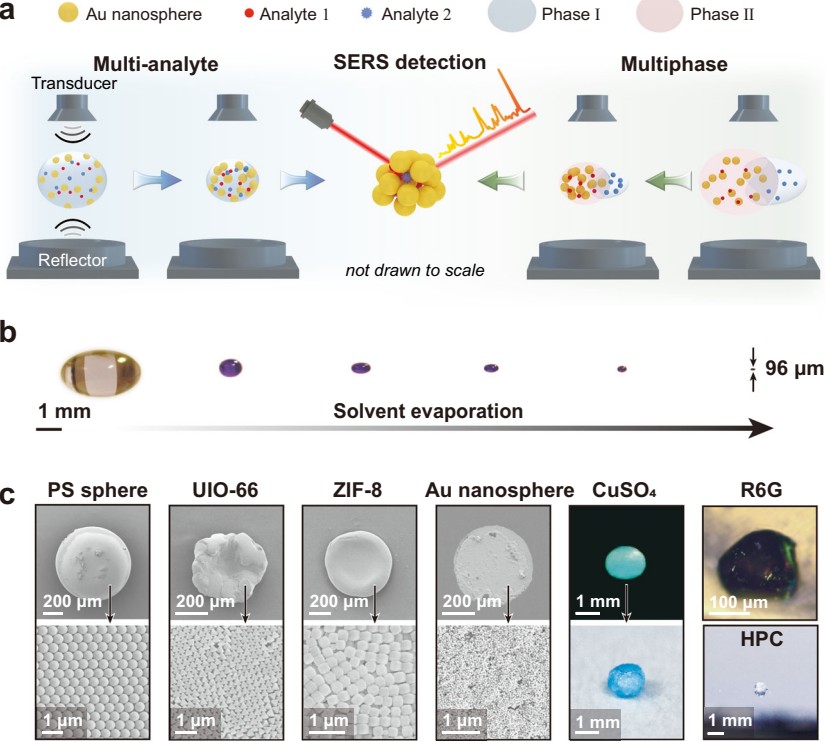

**Fig. 1 | DLE platform enabled multiplex and multiphase SERS detection.**
**a** Schematic of the multianalyte and multiphase enrichment using the DLE platform and the consequent multiplex and multiphase SERS detection. **b** Enrichment process of CV molecules from 10 µL of ethanol solutions. **c** Aggregates concentrated from 10 µL of dispersions of 500 nm polystyrene spheres, 300 nm ZIF-8 octahedra and UIO-66 truncated dodecahedra, and 50 nm Au nanospheres, as well as $CuSO_4$ and R6G aqueous solutions and HPC ethanol solutions.

dodecahedra assembled into concave pancakes after water evaporated. The droplet composed of 50 nm Au nanoparticles was picked up using a piece of silicon wafer from the acoustic levitator when the droplet was almost invisible; the concentration of the Au nanoparticles inside the droplet was significantly increased (Supplementary Fig. 4). A thin film formed by densely packed Au nanoparticles appeared on the silicon wafer. A crystalline $CuSO_4$ sphere was formed after water completely evaporated from a water droplet composed of $CuSO_4$. The weight of the $CuSO_4$ crystal was equivalent to that dissolved in water, proving the lossless enrichment. A rhodamine 6G (R6G) analyte aggregate was formed after water completely evaporated from a levitating R6G ethanol droplet. Importantly, hydroxypropyl cellulose (HPC) ethanol solutions are very sticky, and no solid surfaces can enrich HPC molecules into a small aggregate after solvent evaporation (Supplementary Fig. 3). In contrast, a spherical HPC sphere appeared after ethanol evaporated from a levitating ethanol droplet composed of HPC.

Spherical Au nanoparticles with a size of 50 nm (Fig. 2a) were introduced to the levitating droplet formed by the analyte solutions because of their strong SERS enhancement performance (Supplementary Fig. 5). During solvent evaporation, the Au nanoparticles and analyte molecules were simultaneously

concentrated. A piece of silicon wafer was used to pick up the levitating droplet when its volume was reduced from 10 μL to ~2 nL as more smaller droplets would oscillate in our acoustic levitator and became almost invisible. After a very small amount of solvent evaporated, an aggregate composed of Au nanoparticles and analyte molecules was formed (Fig. 2b). Strong electromagnetic fields (i.e., hot spots) were located at the nanoscale gaps between adjacent Au nanoparticles (Fig. 2c). Two heating lamps were used to accelerate the solvent evaporation process, significantly reducing the evaporation time to less than 30 min (Supplementary Fig. 6). Forcing analyte molecules without interactions with noble metal surfaces into SERS hot spots is challenging for conventional SERS substrates[1,12]. In contrast, both analyte molecules with interactions and those without interactions with Au nanoparticles can be enriched and delivered to SERS hot spots for the DLE platform.

CV molecules were utilized to evaluate the SERS sensitivity of the SERS platform with the DLE function. Ten pM Au nanoparticles were introduced to 10 μL of CV ethanol solutions as too little would make the aggregate difficult to visualize, while too much would decrease the analyte number density within the aggregate (Supplementary Fig. 7). Obvious SERS peaks from CV molecules were observed after

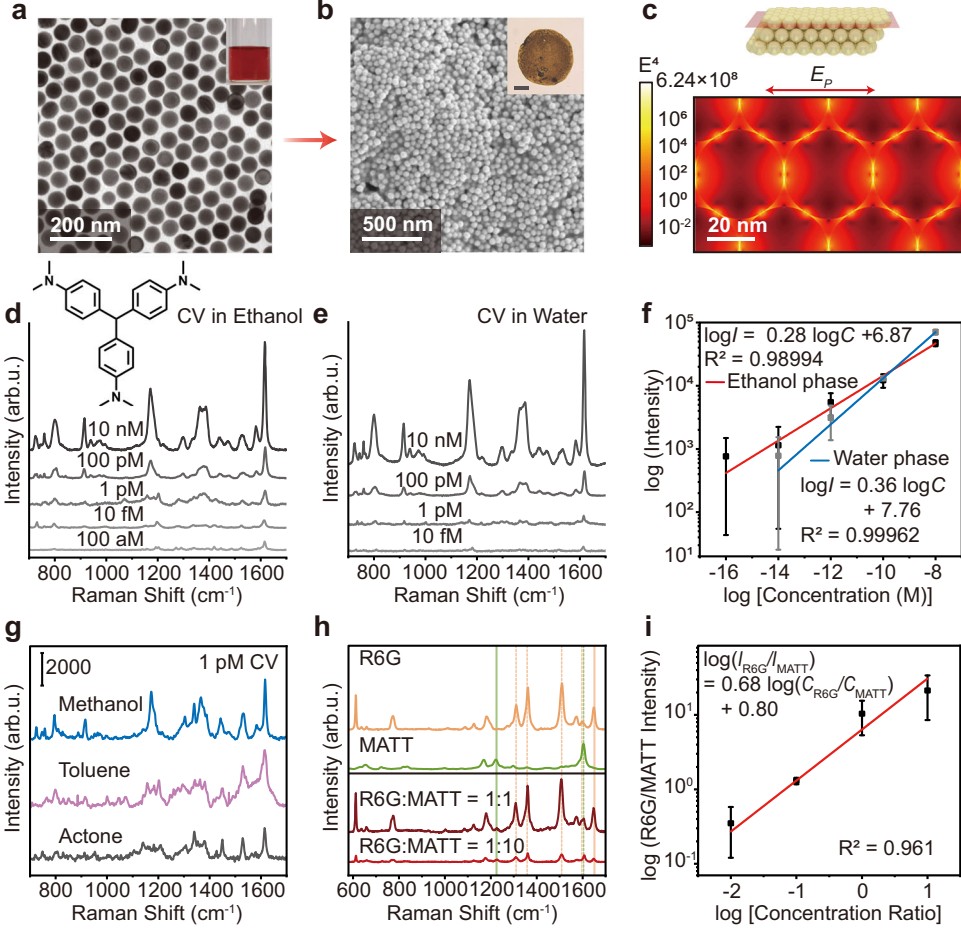

**Fig. 2 | Sensitive SERS detection in various liquids using the DLE platform.**
**a** Transmission electron microscope image of the Au nanospheres. Inset: Photography of the Au nanosphere colloidal dispersion. **b** Simultaneous enrichment of Au nanospheres and CV molecules. Inset: Microscopy image of the Au nanosphere/CV molecule aggregate. Scale bar: 100 μm. **c** FDTD simulated electromagnetic field distribution over the Au nanosphere aggregate. **d, e** SERS spectra of CV molecules concentrated from ethanol solution and aqueous solution, respectively, with different starting concentrations. Inset in **d**: Molecular structure of CV. **f** Relationship

between the intensity of the 1616 cm⁻¹ SERS peak and the concentration of the CV molecules in ethanol and water. The error bars were obtained based on at least 10 spectra. **g** SERS detection of 1 pM CV molecules in methanol, toluene, and acetone. **h** Simultaneous detection of R6G and MATT molecules at different concentration ratios. **i** Relationship between the SERS intensity ratio (1647 cm⁻¹ SERS peak of R6G and 1226 cm⁻¹ SERS peak of MATT) and the concentration ratio ($C_{R6G}/C_{MATT}$). The error bars were obtained based on at least 10 spectra.

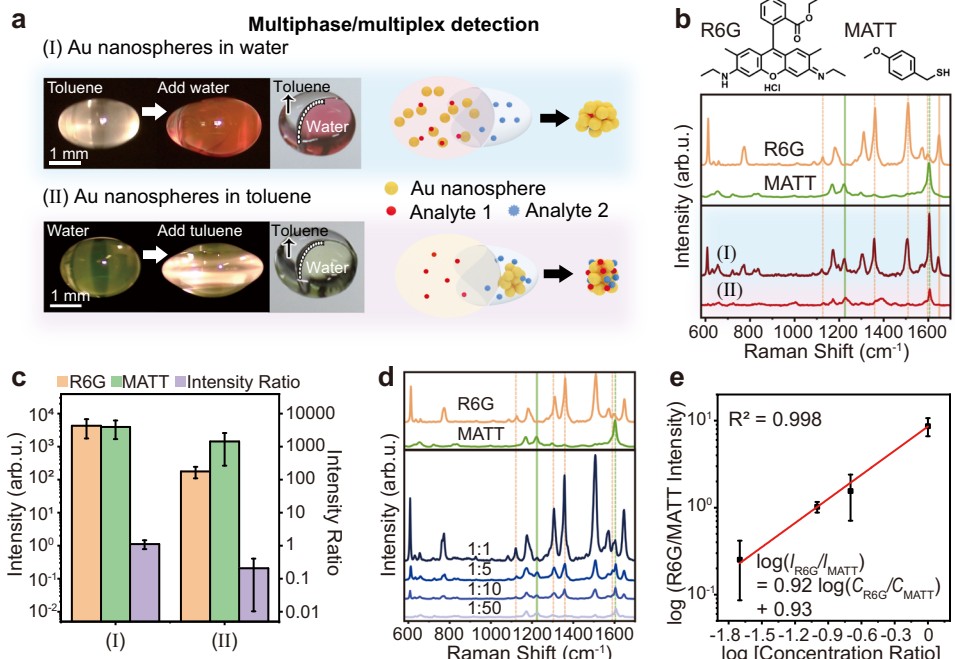

**Fig. 3 | Multiphase and multiplex SERS detection. a** Introducing water into a levitating toluene droplet, or vice versa. Toluene wrapped one end of the ellipsoid water droplet. For multiphase and multiplex SERS detection, introducing Au nanoparticles to the water phase facilitated the analytes to enter the Au nanoparticle aggregate after solvent evaporation; otherwise, the preaggregated Au nanoparticles in the organic phase hindered the entrance of the analytes, particularly for the analytes in the water phase (refer to the schematics on the right). **b** SERS spectra of 50 nM MATT in toluene and 50 nM R6G in water when Au nanoparticles were dispersed in water (Curve I) and aggregated in toluene (Curve II). **c** Intensity of the 1647 cm$^{-1}$ SERS peak of R6G and the 1226 cm$^{-1}$ SERS peak of MATT and their intensity ratio in Curves I and II in panel (**b**). **d** SERS spectra of MATT and R6G at different concentration ratios. **e** Relationship between the intensity ratio of the 1647 cm$^{-1}$ SERS peak of R6G and the 1226 cm$^{-1}$ SERS peak of MATT and the concentration ratio between R6G and MATT in different liquids. The error bars were obtained based on at least 10 spectra.

enrichment from their solutions with a starting concentration in the range of 10 nM to 10 fM composed of 10 pM Au nanospheres using the DLE platform (Fig. 2d). SERS mapping measurements had to be performed to address the location of the CV molecules within the Au nanoparticle/CV molecule aggregate when the starting concentration was 100 aM or lower (Supplementary Fig. 8). The SERS enhancement factor (EF) of the DLE platform was estimated to be ~5.55 × 10$^{13}$ when the starting concentration was 100 aM. The intensity of the SERS signals saturated when the starting concentration was higher than 100 nM as all of the SERS hot spots were occupied by CV molecules after ethanol evaporated (Supplementary Fig. 9). Similarly, CV molecules in aqueous solutions in the concentration range of 10 nM to 100 fM could be detected after enrichment with 40 pM Au nanoparticles using the DLE platform (Fig. 2e). The relationship between the concentration of CV molecules in ethanol/water and the intensity of the 1616 cm$^{-1}$ SERS peak is described by log $I = 0.28$ log $C + 6.87$/log $I = 0.36$ log $C + 7.76$ with an $R^2$ (goodness of fit) of ~ 0.9899/0.9996 (Fig. 2f). The LOD was calculated to be 7.05 aM using Student's t-distribution (details are provided in the "Methods" section), which was at least four orders of magnitude better than conventional SERS substrates without using the DLE technique. CV molecules in other organic solvents with a starting concentration of 1 pM were easily detected (Fig. 2g). In principle, the DLE platform can be employed for ultrasensitive SERS detection in volatile solvents. For biological applications, adenosine and adenine, which are related to life processes, can be easily detected by SERS in aqueous solutions using the SERS platform with the DLE function (Supplementary Fig. 10).

## Multiphase and multiplex trace analyte detection

Multiplex SERS detection was further demonstrated using the DLE platform. SERS spectra of a mixture of R6G and 4-methoxy-α-toluenethiol (MATT) molecules codissolved in ethanol at four

different concentration ratios (i.e., 50 nM:50 nM, 5 nM:50 nM, 500 pM:50 nM, and 50 nM:5 nM) were measured after enrichment using the DLE system (Fig. 2h and Supplementary Fig. 11). The SERS spectra of the mixture could be deconvoluted to those of the two molecules, with the SERS peak intensity dependent on their concentrations. The relationship between the concentration ratio and the SERS intensity ratio is described by log ($I_{R6G}/I_{MATT}$) = 0.68 log ($C_{R6G}/C_{MATT}$) + 0.80, indicating a good quantitative ability in multiplex SERS detection (Fig. 2i).

Multiphase SERS detection using the DLE platform was further demonstrated (Fig. 3). Aqueous solutions can be introduced into the levitating droplet of organic liquids (e.g., toluene), or vice versa (Fig. 3a). Toluene wrapped one side of the ellipsoid water droplet. When Au nanospheres were dispersed in water, the analyte molecules dispersed in both toluene and water could be trapped within the Au nanosphere aggregates after the solvent evaporated. However, Au nanospheres immediately aggregated in toluene, making it difficult for the analyte molecules either dispersed in water or in toluene to enter the Au nanoparticle aggregates after the solvent evaporated. Therefore, strong SERS signals of R6G dispersed in water and MATT dispersed in toluene were simultaneously observed when Au nanoparticles were dispersed in water (Curve I in Fig. 3b). In contrast, the intensity of the SERS signals was reduced by ~2 times for MATT dispersed in toluene and by > 10 times for R6G dispersed in water (Curve II in Fig. 3b). The SERS intensity ratio between R6G and MATT indicated that the R6G molecules dissolved in water had difficulty entering the preformed Au nanoparticle aggregate in toluene (Fig. 3c). SERS spectra of R6G in water and MATT in toluene with different concentration ratios (50 nM:50 nM, 10 nM:50 nM, 5 nM:50 nM, and 1 nM:50 nM) were measured by introducing Au nanoparticles to the water phase after solvent evaporation using the DLE platform (Fig. 3d). The relationship between the concentration ratio

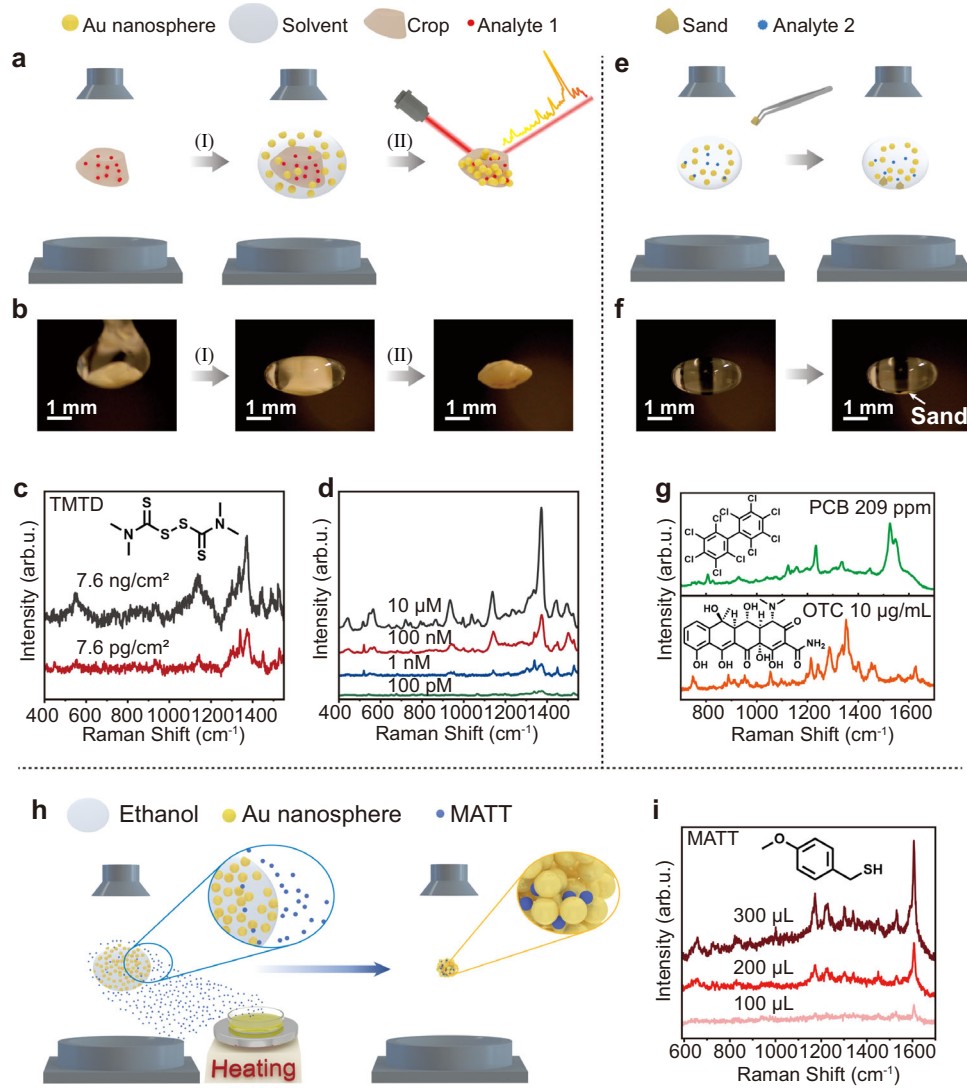

**Fig. 4 | Direct SERS detection of analytes on solid objects and in air using the DLE platform. a, b** Schematic and experimental results, respectively, of direct SERS detection of pesticides on a sesame-seed slice using the DLE platform. Process I in (**b**). Introducing Au nanoparticle colloids onto a levitating sesame-seed slice (long axis: ~2.18 mm). Process II in (**b**). After solvent evaporation, SERS measurements were performed. **c** SERS spectra of TMTD pesticide at different concentrations directly measured from its contaminated sesame-seed slice using the DLE platform. **d** SERS spectra of TMTD from its ethanol solutions at different concentrations using the DLE platform. **e, f** Schematic and experimental results, respectively, of direct SERS detection of environmental pollutants on sand grains using the DLE platform. **g** SERS spectra of PCB 209 and OTC directly measured on sand grains using the DLE platform. **h** Schematic of capturing, enrichment, and SERS detection of airborne molecules using the DLE platform. **i** SERS spectra of MATT molecules captured and enriched from air by an ethanol droplet composed of Au nanospheres. The MATT molecules in air were generated by heating the MATT liquids 10 cm from the levitating ethanol droplet.

and the SERS intensity ratio of R6G and MATT is described by $\log (I_{R6G}/I_{MATT}) = 0.92 \log (C_{R6G}/C_{MATT}) + 0.93$ with an $R^2$ of 0.998 (Fig. 3e).

## SERS detection in the solid phase and gas phase

It is frequently required to detect analytes attached to tiny solid objects (e.g., pesticide residue and soil contamination). Multiple pretreatment processes are usually required to separate analytes from solid objects before detection. The DLE system allowed us to directly analyze the analytes on tiny solid objects by SERS (Fig. 4). Using pesticide residue detection on a slice of sesame seed as an example, the sesame seed slice was levitated by the acoustic levitator (Fig. 4a). Then, 10 μL of ethanol composed of Au nanospheres was dropped onto the seed slice (Process I in Fig. 4a, b). The acoustic force helped peel off the pesticide molecules from the seed slice. As the solvent evaporated, the Au nanoparticles and pesticide molecules simultaneously assembled onto the seed surface (Process II in Fig. 4a, b). Tetramethylthiuram disulfide (TMTD) was selected as a

model pesticide because of its wide usage all over the world. SERS signals of TMTD were observed from the sesame contaminated by TMTD at a concentration of 7.6 pg/cm² without any pretreatment procedures using the DLE platform (Fig. 4c). Sensitive SERS detection of the TMTD ethanol solutions at a concentration as low as 100 pM was realized using the DLE platform (Fig. 4d). We further demonstrated the application of the DLE platform in the SERS detection of environmental pollutants and antibiotics. Decachlorobiphenyl 209 (PCB 209) is a polychlorinated biphenyl (PCB) that can cause serious health issues even at extremely low concentrations[53]. It is difficult to levitate PCB 209-contamined sand grains with a diameter of <0.2 because of their small size, high density, and low power of our acoustic levitator. Therefore, we levitated an ethanol droplet composed of Au nanoparticles. Then, several sand grains were introduced into the ethanol droplet (Fig. 4e, f). After ethanol was evaporated, the SERS signals of PCB 209 were directly observed by measuring a single sand grain (Fig. 4g). Oxytetracycline (OTC) is a

commonly used antibiotic. Similar to the detection process of PCB 209, OTC was also analyzed by direct SERS measurement of a sand grain contaminated by OTC using the DLE platform (Fig. 4g).

SERS detection of airborne analytes is challenging[54,55]. Complex procedures must be followed to capture the analytes spreading in air. We demonstrated efficient capture, enrichment, and SERS detection of airborne analytes using the DLE platform without any peripheral equipment (Fig. 4h, i). Considering that the MATT molecules preferred to dissolve in ethanol, a 10 μL ethanol droplet composed of Au nanoparticles was levitated to trap surrounding MATT molecules spreading in air (Fig. 4h). Once the MATT molecules diffused into the ethanol drop, they anchored onto the Au nanoparticle surface. MATT molecules were enriched and trapped between neighboring Au nanoparticles after ethanol evaporated using the DLE platform, realizing sensitive SERS detection of MATT molecules in air (Fig. 4i). The DLE platform could also be combined with the photoluminescence method to significantly strengthen its sensing capability (Supplementary Fig. 12).

In conclusion, the DLE platform conquers the limitation of the slippery solid surface-based analyte enrichment scenario, allowing us to efficiently concentrate and drive analyte molecules into a localized area in any volatile liquid, on solid objects, or in air. Combining the DLE platform with the SERS sensing technique, we realize multiphase and multiplex analysis at femtomolar levels. The LOD can be further improved simply by increasing the starting volume of the levitating droplet. The DLE platform is compatible with most microsensing techniques. We suggest that the detection performance of traditional analytical methods can be significantly improved by combining them with the DLE platform, potentially expanding their applicability in the fields of biomedicine, environmental science, food safety, and counterterrorism.

## Methods

### Chemicals
Cetyltrimethyl ammonium bromide (CTAB, ≥99%), sodium borohydride (NaBH$_4$, ≥99%), polyvinyl pyrrolidone (PVP, molecular weight 55 kg mol$^{-1}$), and hydroxypropyl cellulose (HPC, 99%) were purchased from Sigma−Aldrich. Chloroauric acid (HAuCl$_4$·3H$_2$O, 99%), crystal violet (CV, >95%), and rhodamine 6G (R6G, >95%) were obtained from Macklin Chemical. Reagent Co., Ltd. Cetyltrimethylammonium chloride (CTAC, >95%) was obtained from Tokyo Chemical Industry Co. Ltd. Ascorbic acid (AA, 99%) was purchased from Alfa Aesar. Adenine (>99%) and adenosine (>99%) were purchased from Sinopharm Chemical Reagent Co., Ltd. 4-Methoxy-α-toluenethiol (MATT, 99%) was obtained from Shanghai Aladdin Biochemical Technology Co., Ltd. All the reagents were used as received. Milli-Q water (>18.2 MΩ·cm at 25 °C) was utilized in all experiments.

### Preparation of Au nanospheres
Single-crystal Au nanospheres with different sizes were synthesized by a seed-mediated growth method[56]. The first step is fabrication of the CTAB-capped Au clusters. HAuCl$_4$ (0.5 mM, 5 mL) and CTAB (200 mM, 5 mL) were ultrasonically and thoroughly mixed. Then, fresh NaBH$_4$ (1 mM, 0.6 mL) aqueous solutions were rapidly added to the above mixture. The solution immediately turned brown. After magnetic stirring for 2 min, the solution was kept undisturbed at 27 °C for 3 h. CTAC (0.2 M, 6 mL), AA (0.1 M, 4.5 mL) and 150 μL of CTAB-capped Au cluster were mixed in a 30 mL glass vial. HAuCl$_4$ solutions (0.5 mM, 6 mL) were then added to the vial by one-shot injection. The mixture was stirred at 27 °C for 15 min. The prepared Au seeds were collected by centrifugation at 28,600 × g for 20 min and washed with water one time. The Au seeds were dispersed in 3 mL of CTAC (0.02 M) for further growth. Aqueous solutions of CTAC (0.1 M, 40 mL), AA (0.01 M, 2.2 mL), and 200 μL of Au seeds was mixed in a conical flask. Then, HAuCl$_4$ (0.5 mM, 40 mL) aqueous solutions were introduced to the flask by a syringe pump at an injection rate of 40 mL/h. The solution was kept at 27 °C for 10 min after the injection finished. The prepared Au nanoparticles were collected by centrifugation at 9500 × g for 15 min and washed with water one time.

### Droplet levitating system
A single-axis acoustic levitator consisting of an acoustic emitter and a reflector was employed (SonaRh-1, Shengdu Ltd., China). The frequency of the acoustic emitter was 20.7 kHz. The operating power was 200 W. Two ceramic heating lamps (-100 W) were utilized to accelerate the droplet evaporation process.

During the droplet levitating process, the sound intensity was slightly controlled by adjusting the distance between the acoustic emitter and the reflector[52]. To levitate an ethanol droplet, we increased the reflector-emitter distance to decrease the sound intensity. Then, an ethanol droplet hanging on a pipette tip was exposed to the acoustic field. We slowly decreased the distance to pull the droplet to leave the pipette tip, obtaining the levitating ethanol droplet. The water droplet was unlikely to explode with suitable sound intensities, and the reflector-emitter distance remained unchanged during the water droplet loading process.

The levitation force per unit volume of an ellipsoid droplet is larger than that on a spherical droplet with the same volume. The aspect ratio of the axes of the levitating ellipsoid droplet should remain at -2. If the ratio is larger than 2, the droplet tends to explode. If the ratio is smaller than 2, the droplet tends to escape from the acoustic field (drops down for large droplets and ejects for tiny droplets). The ellipsoid droplet gradually turns quasi-spherical during evaporation. Slightly reducing the distance between the acoustic emitter and the reflector by -1 mm for our apparatus could squeeze the spherical droplet back to the ellipsoid shape. Air flows should be avoided for levitating very tiny droplets (e.g., <50 nL).

### Preparation of slippery surfaces
The SLIPSs were prepared by pouring a perfluorinated fluid (DuPont Krytox GPL 102) into a porous Teflon membrane[12]. Briefly, a piece of Teflon membrane was attached to a piece of glass slide. Then, the perfluorinated fluid was sprayed onto the Teflon membrane. The polydimethylsiloxane (PDMS) slippery surface was prepared via an acid-catalyzed polycondensation process of dimethyldimethoxysilane[27]. First, a piece of oxygen plasma-treated silicon substrate was immersed into a solution composed of isopropanol (10 g), dimethyldimethoxysilane (1 g), and sulfuric acid (0.1 g) for 10 s. Second, the silicon substrate was dried at room temperature (25 °C, 65% relative humidity) for 30 min before rinsing with isopropanol and water. The hydrophobic perfluorinated surface was fabricated by putting an oxygen plasma-treated silicon substrate and 20 μl of 1H, 1H, 2H, 2H-perfluorodecyltrichlorosilane in a petri dish for 12 h.

### Characterization
The morphology of the aggregates was observed by a scanning electron microscope (Hitachi SU-8010). The shape of the Au nanospheres was characterized by a JEOL JEM-2010 transmission electron microscope operated at 200 kV. UV−Vis absorption spectra of the Au nanosphere colloids were acquired on a UV-2006 spectrometer (Shimadzu, Japan).

### SERS measurements
CV solutions with 10 pM Au nanospheres were obtained by mixing the Au nanosphere colloid and CV solutions. Ten microlitres of the mixture solution was levitated by the single-axis acoustic levitator. As the solvent evaporated, the concentration of the levitating droplet gradually increased. When the diameter of the droplet was less than -150 μm, it started to rapidly oscillate. We transferred the droplet onto a piece of silicon wafer for SERS analysis.

SERS measurements were conducted on a Renishaw inVia confocal Raman microscope spectrometer equipped with a 633 nm laser. A 50× objective was used for all the SERS experiments. The laser power was -0.09 mW. The integration time for CV molecules in ethanol solutions was 1 s. The integration time for other SERS measurements was 10 s, unless otherwise specified. SERS mapping measurements over the aggregates were performed at a step of 1 μm.

## Evaluation of the SERS enhancement factor and LOD

Ten microlitres of 100 aM CV ethanol solution with Au nanospheres was enriched by an acoustic levitator. The intensity of the SERS peak at 1616 cm$^{-1}$ was 3158 counts. Similarly, 10 μL of 1 mM CV ethanol solution without Au nanospheres was concentrated by an acoustic levitator. The intensity of the Raman peak at 1616 cm$^{-1}$ was 569 counts. The SERS EF relying on the DLE platform was evaluated by the following equation:

$$\text{EF} = \left(\frac{I_{\text{SERS}}}{N_{\text{SERS}}}\right) \bigg/ \left(\frac{I_{\text{bulk}}}{N_{\text{bulk}}}\right) \tag{1}$$

where $I_{\text{SERS}}$ is the SERS intensity after enrichment of the 100 aM CV solutions with the Au nanospheres. $I_{\text{bulk}}$ is the Raman intensity after enrichment of 1 mM CV solutions without Au nanospheres. $N_{\text{ads}}$ and $N_{\text{bulk}}$ are the number of CV molecules exposed to the laser spot after enrichment of the 100 aM CV solutions with the Au nanospheres and after enrichment of the 1 mM CV solutions without Au nanospheres, respectively. The SERS EF of the DLE platform was estimated to be -5.55 × 10$^{13}$.

The LOD was estimated using Student's t-distribution[57]:

$$\text{LOD} = \bar{y}_b + t_\alpha^{n-1} s_b^{n-1} \sqrt{\frac{1+1}{n}} \tag{2}$$

where $\bar{y}_b$ is the average blank signal value; $t_\alpha^{n-1}$ is the critical value of the t-distribution; $s_b^{n-1}$ is the standard deviation; and $n-1$ is the degree of freedom, which is 9. Then, we transformed the SERS intensity to the analyte concentration using the relationship between the concentration of CV in ethanol and the SERS intensity of the 1616 cm$^{-1}$ peak[58]:

$$\log I = 0.28 \log C + 6.87 \tag{3}$$

Eventually, the calculated LOD is 7.05 aM.

## FDTD simulations

FDTD simulations were performed using Ansys Lumerical software (Lumerical 2018a). We simulated the electromagnetic field distribution over the Au nanosphere aggregate with an inter-nanoparticle distance of 0.5 nm. The wavelength of the light was 633 nm. The simulation time was 1000 fs.

## Reporting summary

Further information on research design is available in the Nature Portfolio Reporting Summary linked to this article.

# Data availability

All experimental data within the article and its Supplementary Information are available from the corresponding authors upon request.

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

## Acknowledgements
This work was supported by Zhejiang Provincial Natural Science Foundation of China (LR19E010001), the National Key Research and Development Program of China (2018YFB0703803), the National Science Foundation of China (51971200 and 52273233), the China Postdoctoral Science Foundation (512200-X92103), and the Open Research Program of Key Laboratory of 3D Micro/Nano Fabrication and Characterization of Zhejiang Province, Westlake University. Part of the work was conducted in the ZJU micro-nanofabrication center.

## Author contributions
S.Y. conceived the idea. S.Y. and J.R. directed the project. X.C. constructed the experimental setup. X.C. and Q.D. carried out the measurements, analyzed the data, and performed the simulations. C.B. provided testing assistance and analyzed the data. X.C. and S.Y. wrote the manuscript, and all authors participated in discussions.

## Competing interests
The authors declare no competing interests.
