## [Peer Review File · Nature Communications]

Reviewer comments, first round –

Reviewer #1 (Remarks to the Author):

The authors have successfully developed a novel methodology for concentrating analytes in “hot spots” for subsequent SERS analysis. They have been systematic in exploring and optimizing a range of test systems to show the power of the technique using standard very strong Raman scatterers. They also show results on two “biological” chemical analytes adenosine and adenine and go on to show the use of the technique for pesticide residues and contaminants.

The multiplex analyte advantage, which is common to most spectroscopic techniques, is one subject of the paper. It would have been surprising if this was not successful. I doubt whether its inclusion provides anything new.

The multiphase detection results show some limits to what can be achieved using their methodology.

The work is of significance to those working in the SERS field and is original.

One of the issues with the technique is the oscillation of the droplets. This oscillation can become unstable as the drop reduces in size and can lead to ejection of the droplets. The authors should comment on how they control the droplets and stop ejection. What is the minimum drop size in their setup before ejection? Is there success in collecting the concentrated drop on the the silicon slide 100%? If not what is the success rate?

At the end of their conclusion they state that they believe the DLE platform will greatly improve sensitivity for some other analytical techniques. They should support this belief and point out that they would have to be micro-techniques given that they are concentrating to a very small drop size. Ie. Provide some context that supports their “belief”. They also state “The LOD can be further improved simply by increasing the starting volume of the levitating droplet.” What is the maximum drop size that can be levitated in their apparatus and what advantage factor would this give?

For SERS, at some degree of concentration all sites would be covered and subsequent concentration would provide molecules not in “hot regions”. They should explore this limitation.

Background: There are other concentration techniques that have not been included as enrichment methods for SERS. Eg. V shaped grooves, optical trapping.

There are two other Raman based analytical papers that could be considered to put the current work in context. Both aim at directly analysing in the levitated drop. The latter one also demonstrates the use of SERS with acoustic levitation enrichment in situ and so should be included for comparison and to show that the idea of combining SERS and and acoustic levitation has been briefly explored before. The difference to the current work is the extraction of the concentrated drop for later analysis and hence the use of a much smaller diameter laser beam for SERS.

1. Raman acoustic levitation spectroscopy of red blood cells and Plasmodium falciparum trophozoites L Puskar et al Lab on a Chip 2007, 7, 1125 – 1131

2. Chemical Analysis by Raman spectroscopy on acoustically levitated drops, R. Tuckermann et al Anal&Bioanal Chem 394, 1433 – 1441, 2009

Supplementary info.

The coffee ring work was carried out to satisfy themselves. It follows previous works that come to the same conclusions. Simple reference to previous work would suffice unless there is some new approach trying to minimize the problems of the coffee ring effect in what they have done.

For very low concentrations of CV the analytes had to be located by Raman mapping to locate the “hot spots”. Was this necessary in any of the other experiments?

Line 33 “pervaded in air” would be best replaced by “as a gaseous mixture”

Line 173 “sesame” should be “sesame seed”. The measurements of the sesame seed slice should be stated somewhere. Fig 4 shows it

In the experimental section the word “vacillate” should be “oscillate”

Reviewer #2 (Remarks to the Author):

The paper reports on the application of acoustic levitation combined with SERS to detect at low concentrations of analytes using the evaporation generated from the acoustic levitation device to

concentrate both SERS active nano-particles and the analyte of interest. The novelty of the work, from what has previously been reported, lies in the combination of SERS with acoustic levitation, which has some notable advantages compared to traditional colloidal or patterned surfaces to generate SERS enhancement. Furthermore, investigating multiple phases within the droplet is also novel.

Some more comment on the stability of the droplets would be good. For instance, in my experience using an acoustic levitator the drops can become very unstable after time they explode. Was this observed in any of the droplets used in this study and if not how was this avoided?

There are two forms of evaporation going on in these experiments the natural evaporation that occurs under atmospheric conditions and the evaporation induced by the laser. Can the authors comment on the effects of the laser evaporating the drop compared to the air?

There needs to be a proper calculation for the Limit of Detection by averaging blank samples and comparing this with the lowest concentration detected. The authors state limit of detection (LOD) can principally be endlessly pushed down, simply by increasing the volume of the levitating droplet. But you can't increase the droplet size infinitely as the mass and size is limited by the power and the acoustic frequency of the levitator. A good paper on calculating the LoD for a SERS experiment is Methodologies for assessment of limit of detection and limit of identification using surface-enhanced Raman spectroscopy Sensors and Actuators B: Chemical, Volume 207, Part A, 2015, Pages 437-446.

Line 93 remove "was"

Line 94 after the word "after" put "the" water evaporated

Line 99, remove the word "was"

Line 119 change "visible" to "visualise"

Line 153 Can the authors reword the following sentence. Do the authors mean "difficult" instead of "different". I am unsure what the sentence means.

However, Au nanospheres immediately aggregated in toluene, making the analyte molecules whether dispersed in water or in toluene different to enter the Au nanoparticle aggregate after solvent evaporated.

The following referenced should also be cited in the context of multi-analyte detection

Puskar et al. Raman acoustic levitation spectroscopy of red blood cells and Plasmodium falciparum trophozoites, Lab Chip, 2007, 7, 1125-1131

Reviewer #3 (Remarks to the Author):

The paper of X. Chen et al entitled „Droplet levitation enrichment enabled multiphase and multiplex trace analyte detection“ describes an experimental platform for lossless enrichment of molecules dissolved in volatile liquids, attached to solid objects or spreading in air and for using SERS for sensitive detection of analytes.

The presented concept is interesting, unfortunately it is not new at all. The presented data lack scientific depth with respect to validation of the proposed analytical methods. Therefore, neither a truly innovative concept nor sufficiently validated data are presented. Unfortunately, X. Chen and co-authors are also omitting important prior art both with respect to analytical chemistry and SERS in levitated droplets (also in-situ SERS substrate synthesis in levitated droplets had been shown

already in 2003!) but also with respect to sample preparation techniques that are indeed able to avoid the formation of the so called "coffee ring" effect. If that paper would be the first in showing sample preparation and/or SERS in levitated droplets it could be accepted for publication, as it would be introducing a new concept, but this is not the case.

I therefore can not recommend publication of this manuscript, definitely not in a Nature Communications!

The idea of using acoustically levitated droplets for sample preparation in analytical chemistry has been proposed already by E. Welter et al back in 1997: E. Welter, B. Neidhart „Acoustically levitated droplets — a new tool for micro and trace analysis" *Fresenius J Anal Chem* (1997) 357:345-350

The abstract reads as follows: "First experiences with the technique of acoustical levitation of droplets in the field of analytical and atmospheric chemistry are reported. Acoustical levitation enables the contactless handling of solid and liquid microsamples. This avoids adsorption of analyte at and desorption of contaminants from container walls especially for liquid samples. Common experiments of sample preparation procedures were conducted in levitated drops like liquid/liquid extractions, solvent exchange, and analyte enrichment by evaporation of the solvent. A first approach was made to use acoustical levitation for the simulation of atmospheric chemistry situations."

In-situ synthesis of SERS active nanoparticles in levitated droplets and their use for trace analysis has been shown already in 2003. That paper is clearly superior to the concept presented by X. Chen as it shows a way to perform all required analytical sample manipulation, including SERS synthesis and addition of the analyte, in an automated fashion:

N. Leopold, M. Haberkorn, T. Laurell, J. Nilsson, J.R. Baena and J. Frank and B. Lendl "On-line monitoring of airborne chemistry in levitated nano-droplets. In-situ synthesis and application of SERS active Ag-sols for trace analysis by Raman spectrometry" *Anal. Chem.* 75 (2003) 2166-2171
The abstract reads as follows: "We report a new strategy for on-line monitoring of chemical reactions in ultrasonically levitated, nanoliter-sized droplets by Raman spectroscopy. A flow-through microdispenser connected to an automated flow injection system was used to dose picoliter droplets into the node of an ultrasonic trap. Taking advantage of the flow-through characteristics of the microdispenser and the versatility of the automated flow system, a well-defined sequence of reagents could be injected via the microdispenser into the levitated droplet placed in the focus of the collection optics of the Fourier transform Raman spectrometer. In that way, chemical reactions could be carried out and monitored on-line. The developed system was used for fast, reproducible, in situ synthesis of a highly active surface enhanced Raman scattering (SERS) sol resulting from the reduction of silver nitrate with hydroxylamine hydrochloride in basic conditions. With this chemical system, SERS substrate preparation could be achieved at room temperature and in short time. The in situ prepared silver sol was used for trace analysis of several organic test molecules that were injected into the levitated SERS-active droplet again using the microdispenser. The concentration dependence of the SERS spectra was studied using 9-aminoacridine, revealing that down to the femtogram region high-quality SERS spectra could be obtained. Additionally, SERS spectra of 6-mercaptopurine, thiamine, and acridine were recorded in the levitated drop as well."

With respect to the "coffee ring" effect X. Chen only cites his own work leading to the conclusion that the coffee ring effect can not be avoided. However, the group of Gerwert in Bochum, Germany had proposed a solution to this problem already in 2013:

Julian Ollesch, Steffen L. Drees, H. Michael Heise, Thomas Behrens, Thomas Bruening and Klaus Gerwert „FTIR spectroscopy of biofluids revisited: an automated approach to spectral biomarker identification" *Analyst*, 2013, 138, 4092

In their paper they write: „A compact, benchtop sized robotic spotting system (Instrument2, M2 Automation GmbH, Berlin, Germany) was used to dispense the samples in quadruplicate on 384-well silicon multi-well titerplate (MTP) substrates (Bruker Optics, Ettlingen, Germany). Substrates were cleaned with sodium hypochlorite solution and plasma treatment (Zepto, Diener plasma surface technology, Ebhausen, Germany) before use. Each well covers a circular area of 4 mm diameter, which was almost perfectly covered with the sample. A total volume of 3 ml was loaded at a syringe-pump controlled speed into the dispenser. The single sample lm was formed from

approximately 50 nl. Subsequently, the sample loaded substrate plates were vacuum-dried for 10 min."

They continue later on:

„Due to the extremely low volumes spotted, the formation of a coffee ring as still observed with nanoliter volumes was avoided. An atomic force microscopic (AFM) scan confirmed that at such a low volume, an indented cylinder shape is formed instead of an irregular ring structure (Fig. 5A and B)."

Responses to Referees' comment

Reviewer #1:

Overall Comment: *The authors have successfully developed a novel methodology for concentrating analytes in “hot spots” for subsequent SERS analysis. They have been systematic in exploring and optimizing a range of test systems to show the power of the technique using standard very strong Raman scatterers. They also show results on two “biological” chemical analytes adenosine and adenine and go on to show the use of the technique for pesticide residues and contaminants. The multiplex analyte advantage, which is common to most spectroscopic techniques, is one subject of the paper. It would have been surprising if this was not successful. I doubt whether its inclusion provides anything new. The multiphase detection results show some limits to what can be achieved using their methodology. The work is of significance to those working in the SERS field and is original.*

Author response: We greatly appreciate for your positive and constructive comments, which are all valuable and extremely helpful for improving the quality of our work. We have studied and addressed your comments carefully and revised the manuscript according to your comments/suggestions. The multiplex analyte detection in a single phase can be easily achieved using the SERS technique. Here, we demonstrated multiplex detection in not only a single phase (*e.g.*, water), but also in two immiscible phases (*e.g.*, one analyte in water and the other one in toluene). Multiplex analyte detection in immiscible phases is not easy to achieve without using the analyte enrichment acoustic levitation platform. For multiphase detection, we revealed that the Au nanoparticles should be introduced into the phase where they could be well-dispersed. Otherwise, the aggregated Au nanoparticles would prohibit the analyte molecules to enter the Au nanoparticle aggregates. We are working on fabrication of high performance microscale SERS substrates with nanopores. Using these pre-fabricated SERS microsensors can overcome the above phase-relying limitation. Overall, we demonstrated multiplex analyte detection in single or multiple phases. For

multiphase detection, we explored the experimental conditions that can achieve the best SERS sensing performance. Thank you again for your positive comments.

Comment 1: *One of the issues with the technique is the oscillation of the droplets. This oscillation can become unstable as the drop reduces in size and can lead to ejection of the droplets. The authors should comment on how they control the droplets and stop ejection. What is the minimum drop size in their setup before ejection? Is there success in collecting the concentrated drop on the silicon slide 100%? If not what is the success rate?*

Response 1: Thank you for your comments. The original levitated droplet with a volume of 10 μl kept still in a shape of oblate spheroid. However, along with the evaporation of the droplet, it started to oscillate. When the volume of the droplet reached $\sim 2.5 \mu\text{l}$, the droplet evolved into a quasi-sphere. The quasi-spherical droplet would eject from the acoustic field at any time. According to the theoretical studies (*Physical Review E*, 2004, 70, 046611), the levitation force per unit volume of an oblate spheroid of the droplet is larger than that on the sphere with the same volume in the acoustic field. Therefore, when the droplet reached a quasi-spherical shape, when the acoustic levitation force was barely enough to balance the gravity force, the droplet started to oscillate in a horizontal plane. When the droplet size was further decreased, the acoustic levitation force was not enough to balance the gravity force and the droplet would escape the acoustic field.

Based on the above analysis, it is important to maintain the oblate spheroid shape of the droplet to avoid its ejection from the acoustic field. In other words, we need to intermittently increase the sound intensity to maintain the evaporating droplet in an oblate spheroid shape. For our apparatus, the power (we used 200 W) could not be varied once the droplet levitation started. Adjusting the distance between the reflector and the acoustic emitter can tailor the sound intensity in a small range. Experimentally, we observed the droplet changed from a quasi-sphere to an oblate spheroid when we reduced the emitter-reflector distance. Therefore, we adjusted the emitter-reflector distance to maintain the levitation of the evaporating droplet.

In our experiment, the aspect ratio of the axes of the levitating droplet in the oblate spheroid shape was always kept at ~ 2 by adjusting the reflector-emitter distance. We only need to adjust the reflector-emitter distance twice to maintain the droplet levitating during the evaporation process from $10\ \mu\text{l}$ to $\sim 30\ \text{nl}$. The levitating droplet has a strong tolerance to the moving distance and each time we moved the reflector about $1\ \text{mm}$ upward when we saw the droplet turned to be a quasi-sphere. When the droplet further evaporated from $30\ \text{nl}$, we need to avoid surrounding environment disturbances. Slight air flow may blow away the tiny levitating droplet. The above details about the droplet levitating process have been included in the revised manuscript.

The minimum drop size before ejection for our apparatus is approximately $0.5\ \text{nl}$ (axis: $< 100\ \mu\text{m}$). Since it is hard to distinguish the object with a size less than $100\ \mu\text{m}$, the appropriate drop size suitable for SERS detection is about $2\ \text{nl}$. Therefore, when the droplet was evaporated into $2\ \text{nl}$, we transferred it onto a substrate from the acoustic field for SERS detections.

It is important to reduce the blocking of the acoustic wave by the tweezer and the substrate during the droplet transferring process. We used a sharp stainless steel tweezer to hold the silicon slide ($1\ \text{cm} \times 1.5\ \text{cm}$) to pick up the levitating droplet. The successful transferring rate of the tiny levitating droplet is dependent on the droplet size. When the droplet volume is larger than $10\ \text{nl}$, the successful transferring rate is almost 100% . When the droplet volume is smaller than $10\ \text{nl}$, the successful transferring rate decreases with the decrease of the droplet volume, induced by the oscillation of the tiny droplet. When the droplet volume is $\sim 2\ \text{nl}$ (the volume we used for SERS detection), the successful transferring rate is about 70% . We are now trying to adhere a vertical ultra-slippery fiber ($\sim 100\ \mu\text{m}$ in diameter) onto the reflector to fix the droplet during evaporation. By doing this, the analytes and the Au nanoparticles can be enriched onto a small area on the fiber. The fiber can be directly used for the SERS measurement.

When detect the solid object (*e.g.*, sesame seed slice and sand powder), a piece of copper net (200 mesh) was put on the reflector before levitating the object (the copper net was almost transparent to the sound wave). After applying the Au nanoparticle

colloidal solution onto the levitating solid object and drying, the solid object covered by Au nanoparticles could be 100% trapped by the copper meth.

We have added the above transferring rate discussions in the revised manuscript.

Comment 2: *At the end of their conclusion they state that they believe the DLE platform will greatly improve sensitivity for some other analytical techniques. They should support this belief and point out that they would have to be micro-techniques given that they are concentrating to a very small drop size. Ie. Provide some context that supports their “belief”. They also state “The LOD can be further improved simply by increasing the starting volume of the levitating droplet.” What is the maximum drop size that can be levitated in their apparatus and what advantage factor would this give?*

Response 2: Thank you for your comments. As you said, the DLE platform can concentrate analytes into a small area. Therefore, the DLE platform is only suitable to micro-techniques. We have pointed out this in the revised manuscript. To support the claim that the DLE platform will improve the sensitive of other micro-techniques, we emphasize the importance of the starting concentration and the spreading area of the analytes to the sensitivity in the revised manuscript. Moreover, we took photoluminescence (PL) sensing technique as another example to demonstrate the importance of the DLE platform regarding improving its sensitivity. The PL spectra of 1 fM Rhodamine 6G (R6G) before and after the DLE enrichment was shown in Fig. R1a. Before DLE enrichment, no PL peak was observed. After DLE enrichment, a strong PL peak was observed. The excitation and emission wavelengths were 532 nm and 555 nm, respectively. The fluorescence image captured by a fluorescence microscope clearly showed the aggregate of R6G molecules formed on the DLE platform (Fig. R1b). We added the above contents and figures in the revised Supporting Information.

Fig. R1. The PL enhancement of R6G molecules after enrichment using the DLE platform. a, PL spectra of 1 fM R6G ethanol solution before and after analyte enrichment using the DLE platform. b, The R6G aggregate formed after evaporating ethanol from the levitating droplet observed under a microscope (top) and a fluorescence microscope (bottom). Excitation wavelength: 488 nm.

When the wavelength of the ultrasound exceeds 8 mm, the maximum volume of a liquid drop that can be levitated was determined by the ratio of hydrostatic pressure and capillary pressure due to the surface tension of the drop (*Fresenius J Anal Chem*, 1997, 357:345-350). The frequency of the emitter for our apparatus was 20.7 kHz (wavelength \approx 16.4 mm). The volume of the droplet that can be levitated is therefore determined by the surface tension of the liquid.

According to our experiments, the maximum droplet volume using our apparatus operating at 200 W is about 180 μ l for water and 30 μ l for ethanol (**Fig. R2**). After enrichment by the DLE platform, the expected analyte concentrating times could reach \sim 360, 000 and \sim 60,000 for the aqueous and ethanol droplet, respectively.

Using other acoustic levitators with a longer acoustic wavelength and higher power, even larger droplets should be able to be levitated, and in turn the advantage factor could be further improved.

Fig. R2. The largest droplet of water a, and ethanol b, that can be levitated using our apparatus.

Comment 3: For SERS, at some degree of concentration all sites would be covered and subsequent concentration would provide molecules not in “hot regions”. They should explore this limitation.

Response 3: Thank you for your comments. Crystal violet (CV) dye molecules were used to explore this critical saturation concentration. When the distance between adjacent Au nanospheres is less than 2 nm, the electric field intensity is significantly enhanced (Fig. R3a). We assume that the “hot spots” (same as “hot regions”) basically are covered by CV molecules when the thickness of the molecule layer around each Au nanospheres is 1 nm. 10 pM Au nanoparticles are introduced into 10 μl of CV ethanol solutions. The total volume (V_1) of the molecule layer is $4.73 \times 10^{-7} \text{ mm}^3$. The volume of the smallest rectangular tank of a single CV molecule is about 1.63 nm^3 (Fig. R3b) (*Journal of Physical Chemistry A*, 2009, 113, 5806–5812). The estimated volume (V_2) of two CV molecules is about 1.63 nm^3 because the molecules are packed tightly. The saturation concentration of CV molecules is estimated by $c = 2V_1 / V_2 / N_A / 10 \mu\text{l} = 9.64 \times 10^{-8} \text{ mol/l}$.

Fig. R3. The characterization of hot spots and CV molecule. **a**, FDTD simulated electric field distribution over the Au nanosphere aggregate. **b**, The size of single CV molecule.

We further performed experiments to compare with the above estimated results. The experimental results showed that the saturation concentration was approximately 1×10^{-7} mol/l, which was in good agreement with the calculation value (Fig. R4a). The signal to noise ratio and the SERS intensity of $10 \mu\text{M}$ CV is lower than 100 nM CV (Fig. R4b). It is probably because the crystallization of CV molecules at high concentrations and therefore many CV molecules are not within the “hot spots”. We added the above contents and figures in the revised Supporting Information.

Fig. R4. The saturation concentration for the SERS sensors. **a**, The SERS intensity variation at 1616 cm^{-1} as a function of CV concentrations. **b**, The SERS spectra of CV molecules at $10 \mu\text{M}$ and 100 nM concentrations.

Comment 4: *Background: There are other concentration techniques that have not been included as enrichment methods for SERS. Eg. V shaped grooves, optical trapping.*

Response 4: Thank you for your valuable comments. We have included other analyte enrichment for SERS detections in the revised manuscript. For example, a single evanescent-wave optical excitation was used to achieve plasmonic assembly and ultra-sensitive SERS detection simultaneously (*Nature Communications*, 2014, 5, 4357).

The nanosponge consists of porous β -cyclodextrin polymers immobilized with magnetic nanoparticles could adsorb and enrich pollutants by ~ 1000 times for SERS detections (*Nature Communications*, 2021, 12, 6849).

The geometric design of V-shaped microchannels also enables a “trap” to the molecule confinement and builds up an excellent electromagnetic field distribution by AuNR aggregates (*ACS Applied Materials & Interfaces*, 2021, 13, 36482–36491).

These important references have been added in the revised manuscript.

Comment 5: *There are two other Raman based analytical papers that could be considered to put the current work in context. Both aim at directly analysing in the levitated drop. The latter one also demonstrates the use of SERS with acoustic levitation enrichment in situ and so should be included for comparison and to show that the idea of combining SERS and acoustic levitation has been briefly explored before. The difference to the current work is the extraction of the concentrated drop for later analysis and hence the use of a much smaller diameter laser beam for SERS.*

1. *Raman acoustic levitation spectroscopy of red blood cells and Plasmodium falciparum trophozoites, L Puskar et al, Lab on a Chip 2007, 7, 1125 – 1131*

2. *Chemical Analysis by Raman spectroscopy on acoustically levitated drops, R. Tuckermann et al, Anal&Bioanal Chem 394, 1433 – 1441, 2009*

Response 5: Thank you for your valuable information and suggestions. The references you mentioned are really very helpful and we have cited the above two important papers in the revised manuscript.

The second paper used Raman acoustic levitation spectroscopy (RALS) to study the chemical and physicochemical processes, e.g., concentration series, phase transitions,

and chemical reactions. We have appreciated the combination of SERS and acoustic levitations reported in this reference in the revised manuscript.

Comment 6: *Supplementary info.*

The coffee ring work was carried out to satisfy themselves. It follows previous works that come to the same conclusions. Simple reference to previous work would suffice unless there is some new approach trying to minimize the problems of the coffee ring effect in what they have done.

Response6: Thank you for your suggestions. According to your suggestion, we have deleted Figure S1 associated with the coffee ring work and instead added some related references in the revised manuscript and Supplementary Information.

Comment 7: *For very low concentrations of CV the analytes had to be located by Raman mapping to locate the “hot spots”. Was this necessary in any of the other experiments?*

Response 7: Thank you for your comments. When the concentration of analyte decreases, it is highly possible that there are very few or even no analyte molecules under the laser spot area (several micrometers in diameter). Moreover, the probability of the very few amount of molecules locate at the “hot spots” becomes very low. Therefore, we need to perform SERS mapping to cover a large area to address the molecules. Generally, when the starting concentration is at fM concentration level, we have to perform the SERS mapping measurements.

Comment 8: *Line 33 “pervaded in air” would be best replaced by “as a gaseous mixture” Line 173 “sesame” should be “sesame seed”. The measurements of the sesame seed slice should be stated somewhere. Fig 4 shows it in the experimental section the word “vacillate” should be “oscillate”*

Response 8: Thank you for your comments. We have replaced these errors/typos you mentioned in the revised manuscript. We added the measurements of the sesame seed in the caption of Figure 4 in the revised manuscript.

Reviewer #2 (Remarks to the Author):

Overall Comment: *The paper reports on the application of acoustic levitation combined with SERS to detect at low concentrations of analytes using the evaporation generated from the acoustic levitation device to concentrate both SERS active nanoparticles and the analyte of interest. The novelty of the work, from what has previously been reported, lies in the combination of SERS with acoustic levitation, which has some notable advantages compared to traditional colloidal or patterned surfaces to generate SERS enhancement. Furthermore, investigating multiple phases within the droplet is also novel.*

Author response: We are very grateful for your positive comments and valuable suggestions, which have greatly improved the quality of our manuscript. We have addressed your comments and revised the manuscript according to your suggestions.

Comment 1: *Some more comment on the stability of the droplets would be good. For instance, in my experience using an acoustic levitator the drops can become very unstable after time they explode. Was this observed in any of the droplets used in this study and if not how was this avoided?*

Response 1: Thank you for your comments. Explosion of the levitating droplet was also observed in our experiment, especially for the ethanol droplet. To understand and solve the droplet explosion issue, we analyzed the cause of the explosion. When the wavelength of the ultrasound exceeds 8 mm, the maximum volume of a liquid droplet that can be levitated is determined by the ratio of the hydrostatic pressure and the capillary pressure due to the surface tension of the liquid. If this ratio, namely the Bond number of the levitated droplet, exceeds 1.5, the droplet disintegrates and will explode (*Fresenius J Anal Chem*, 1997, 357:345-350). The frequency of the acoustic emitter in our work was 20.7 kHz (wavelength \approx 16.4 mm). Because of the lower surface tension of ethanol (22.3 mN/m at 20 °C) than that of water (72.8 mN/m at 20 °C), the ethanol droplet is easier to explode than water droplet.

Since it is not easy to greatly change the surface tension of a liquid, we try to lower the hydrostatic pressure. Aqueous or organic liquid can be levitated without explosion by tuning the sound radiation pressure according to their different surface tension. We found that the radiation pressure generated by our acoustic levitator operating at 200 W was appropriate. This power generated enough force to hold the droplet without over-squeezing the droplet (the over squeezing is one main reason caused the explosion). The shape of the liquid droplet is being continuously changed from a quasi-sphere to an oblate spheroid when the sound intensity increases (*Physical Review E*, 2004, 70, 046611). The sound intensity increase was enabled by reducing the distance between the sound emitter and the reflector (the power could not be adjusted once fixed at the beginning for our apparatus). To introduce the ethanol droplet into the DLE platform, we first increased the reflector-emitter distance to decrease the sound intensity. Then the ethanol droplet was exposed to the acoustic field hanging on a pipette tip. We slowly decreased the distance to pull the droplet leave the pipette tip. The largest ethanol droplet that can be levitated is 30 μl for our apparatus. The analyte enrichment factor can read ~ 60000 for the ethanol droplet. By increasing the wavelength of the acoustic wave, larger ethanol droplet can be levitated. The water droplet is unlikely to explode under suitable sound intensity and the reflector-emitter distance could remain unchanged during the water droplet loading. The largest water droplet that can be levitated is 180 μl for our apparatus. The analyte enrichment factor can read ~ 360000 for the water droplet.

The levitation force per unit volume of an oblate spheroid of the droplet is larger than that on the sphere with the same volume. Our experience is that the aspect ratio of the axes of the levitating droplet should keep at ~ 2 in a shape of oblate spheroid. Larger than 2, the droplet tends to explode. Smaller than 2, the droplet tends to escape from the acoustic field (drop down for large droplet and eject for the tiny droplet).

We have added the experimental details to avoid the explosion of the droplet in the revised Supporting Information.

Comment 2: *There are two forms of evaporation going on in these experiments the natural evaporation that occurs under atmospheric conditions and the evaporation induced by the laser. Can the authors comment on the effects of the laser evaporating the drop compared to the air?*

Response 2: Thank you for your comments. We are sorry for the confusion. We used two heating lamps to accelerate the evaporating process of the droplet (Fig. R1). We have clearly described that two ceramic heating lamps were used to accelerate the droplet evaporation process in the revised manuscript.

Fig. R1. The heating equipment. Two ceramic heat lamps (100 W, diameter \approx 72 mm).

The heating lamps could increase the temperature of the levitating droplet from 15.8 °C to 30.3 °C in two minutes. The air temperature around the droplet increased from 22 °C to 55 °C (Fig. R2). The relative humidity was 55~60%. The heating lamps increased the evaporation rate by reducing the humidity around the droplet and increasing the temperature of it.

A droplet with a volume of 10 μl needs more than two hours to evaporate in the air (temperature ≈ 22 °C, humidity $\approx 55\%$). When using the heating equipment, the ethanol and water droplet need 28 min and 33 min to evaporate. When the ambient temperature is higher and the humidity is lower, for example, 35 °C and 40%; the evaporation times can be reduced to 18 min (ethanol droplet) and 27 min (water droplet). We are now manufacturing a microwave heating setup, aiming at reducing the evaporation time to less than 5 min. We added the above discussions and figures in the revised Supporting Information.

Comment 3: *There needs to be a proper calculation for the Limit of Detection by*

Fig. R2. The heating lamps increased the temperature of the levitating droplet and accelerated the solvent evaporation process. (a-d) The temperature of the 10 μl of 0.4 nM Au nanosphere colloids increased from 15.8 °C to 30.3 °C in two minutes under the heating lamp irradiation. The air around the droplet is 55 °C. Room temperature: 22 °C.

averaging blank samples and comparing this with the lowest concentration detected. The authors state limit of detection (LOD) can principally be endlessly pushed down, simply by increasing the volume of the levitating droplet. But you can't increase the

droplet size infinitely as the mass and size is limited by the power and the acoustic frequency of the levitator. A good paper on calculating the LOD for a SERS experiment is *Methodologies for assessment of limit of detection and limit of identification using surface-enhanced Raman spectroscopy Sensors and Actuators B: Chemical, Volume 207, Part A, 2015, Pages 437-446.*

Response 3: Thank you for your valuable comments/suggestions. The sentence “The limit of detection (LOD) can principally be endlessly pushed down, simply by increasing the volume of the levitating droplet” has been revised to “The limit of detection (LOD) can be further pushed down simply by increasing the volume of the levitating droplet”.

The reference you referred including extensively used LOD calculation methods in analytical chemistry. In our DLE-SERS sensing system, we could increase the analyte concentration by evaporating the solvent. The concentration enrichment could reach thousands of times.

According to your suggestion, we chose one of the approaches to estimate the LOD of our DLE-SERS platform using the Student’s t-distribution (*Sensors and Actuators B, 2015, 207, 437–446*).

$$\text{LOD}_B = \bar{y}_b + t_{\alpha}^{n-1} s_b^{n-1} \sqrt{\frac{1+1}{n}} \quad (1)$$

Here the \bar{y}_b is the average blank signal value, t_{α}^{n-1} is the critical value of the t-distribution, s_b^{n-1} is the standard deviation, $n - 1$ is the degree of freedom and here is 9. The transformed data from SERS signal intensity to the analyte concentration using the following equation (*Spectrochimica Acta Part A: Molecular and Biomolecular Spectroscopy, 2013, 111, 237–241*):

$$\log I = 0.28 \log C + 6.87 \quad (2)$$

The above equation is the relationship between the concentration of CV molecules in ethanol and the SERS intensity of the 1616 cm^{-1} peak. The calculated value of LOD_B is 7.05 aM for our DLE-SERS platform.

Your comments have greatly improved the quality of our work. We have refined the above contents and added them into our revised manuscript.

Comment 4: *Line 93 remove “was”*

Line 94 after the word “after” put “the” water evaporated

Line 99, remove the word “was”

Line 119 change “visible” to “visualise”

Line 153 Can the authors reword the following sentence. Do the authors mean “difficult” instead of “different”. I am unsure what the sentence means. “However, Au nanospheres immediately aggregated in toluene, making the analyte molecules whether dispersed in water or in toluene different to enter the Au nanoparticle aggregate after solvent evaporated.”

Response 4: Thank you for your valuable suggestions. We have revised the grammar errors according to your comments in the revised manuscript. As you said, “different” should be “difficult”. We are sorry for this typo.

Comment 5: *The following referenced should also be cited in the context of multi-analyte detection: Puskar et al. Raman acoustic levitation spectroscopy of red blood cells and Plasmodium falciparum trophozoites, Lab Chip, 2007, 7, 1125–1131*

Response 5: Thank you for your suggestions. The important reference you mentioned is really helpful and we have cited the above paper in the revised manuscript.

Reviewer #3 (Remarks to the Author):

Overall Comment: *The paper of X. Chen et al entitled, Droplet levitation enrichment enabled multiphase and multiplex trace analyte detection “describes an experimental platform for lossless enrichment of molecules dissolved in volatile liquids, attached to solid objects or spreading in air and for using SERS for sensitive detection of analytes. The presented concept is interesting, unfortunately it is not new at all. The presented data lack scientific depth with respect to validation of the proposed analytical methods. Therefore, neither a truly innovative concept nor sufficiently validated data are presented. Unfortunately, X. Chen and co-authors are also omitting important prior art both with respect to analytical chemistry and SERS in levitated droplets (also in-situ SERS substrate synthesis in levitated droplets had been shown already in 2003!) but also with respect to sample preparation techniques that are indeed able to avoid the formation of the so called “coffee ring” effect. If that paper would be the first in showing sample preparation and/or SERS in levitated droplets it could be accepted for publication, as it would be introducing a new concept, but this is not the case. I therefore can not recommend publication of this manuscript, definitely not in a Nature Communications!*

Author response: Thank you for your comments and constructive suggestions. As you said, combining SERS and droplet levitation is not a new idea. Dozens of papers have used the SERS technique to analyze the composition of a levitating droplet or monitor the crystallization/reaction processes within a levitating droplet. The first report on the combination of the SERS technique and acoustic levitation should be around 2003 by Prof. Lendl's group (Using SERS to monitor chemical reactions within a levitating droplet published on *Anal. Chem.* 2003, 75, 2166; Using SERS monitoring the crystallization process within a levitating droplet published on *Anal. Chem.* 2003, 75, 2177). Prof. Lendl's group further performed many brilliant studies on SERS analysis of a levitating droplet. We have appreciated Prof. Lendl's pioneering work on the combination of SERS and acoustic levitation in our revised manuscript. The 2003 paper (*Anal. Chem.* 2003, 75, 2166) developed a unique automated sample loading micro-

dispenser system to the acoustic levitator, enabling different chemical reactions to take place and to be monitored on-line by the SERS technique. Silver sol solutions were prepared and used for trace analysis of several organic test molecules.

Different from the 2003 paper and other previous studies, the novelty of our work lies in analyte enrichment within a levitating droplet by evaporating the solvent for ultrasensitive SERS detection. The concentration could be increased by more than 20000 times using a 10 μ l levitating droplet, which is different to achieve using other methods. We also demonstrated multiphase detection using the levitating droplet composed of water and organic liquids. Multiphase SERS detection is challenging (see *Nat. Mater.* 2013, 12, 165-171), which can be easily achieved after using the acoustic levitating droplet. Moreover, we showed *in situ* SERS analysis of solid objects (*e.g.*, sesame seed slice and sand powder) by applying Au nanoparticle colloidal solutions onto the levitating solid objects. After ethanol evaporated, SERS analyses of the solid objects could be directly performed. We further demonstrated concentrating and SERS detection of airborne molecules using a levitating droplet composed of Au nanoparticles to absorb the molecules in air. The above contents have never been reported before. The analyte enrichment using an acoustically levitated droplet for SERS detections will attract broad research interest from the acoustic levitator and SERS community.

The inspiration of using the levitating droplet to perform SERS measurements comes from our previous investigations. Many studies have carried out to design SERS substrates with uniformly distributed SERS “hot spots” (sensitive SERS sites) to increase the reproducibility and sensitivity of SERS. Control the spatial distribution of analytes on the SERS substrates is also important to attain the above purpose, however have been rarely studied \sim 10 years ago. Generally, the analyte molecules will form a large “coffee ring” (\sim 1 cm in diameter) with more molecules concentrated at the edge than in the center area. Therefore, the SERS reproducibility and sensitivity vary at different locations of the SERS substrates. We first introduced slippery liquid-infused surfaces (SLIPS) that can repel almost any liquids into the SERS system (referred as SLIPSERS), enabling to concentrate the analytes into a tiny dot (\sim several hundred

micrometers in diameter) from almost any volatile solvents after evaporation. We would like to enrich the analytes into micrometer area not only because the concentrated analyte molecules more evenly distributed on the SERS substrate, but also *the size of the laser spot used for SERS detection is in this size range*. The analytes that cannot be covered by the small laser spot cannot contribute to the SERS signals. The concentration enrichment efficiency of SLIPSERS reached several thousand times and improved the SERS detection limit by at least three orders (*PNAS*, 2016, 113, 268-273; *Nano Letters* 2020, 20, 7304-7312).

As mentioned above, we would like to enrich the analytes from highly diluted solutions into a small dot (ideally 100 μm in diameter considering this is the smallest size that naked eyes can observe and it is only slightly larger than the laser spot size of sever micrometers) with evenly distributed analytes. The analyte concentration increased thousands of times from the droplet to the small dot on the slippery surfaces. *The paper you mentioned about overcoming the coffee ring effect used a very small drop and an analyte aggregate with a flat top surface was obtained after solvent evaporation. However, the drop did not shrink during the solvent evaporation, and therefore loss the analyte concentration function that highly desired for improving the SERS sensitivity.*

We found that a small amount of analytes actually failed to be concentrated into the tiny spot due to the wetting defects or the contamination of the existing slippery surfaces. Some carefully designed slippery surfaces may be able to completely avoid the coffee ring effect for some liquids, but it is difficult to design surfaces that can avoid the coffee ring effect for any liquids, particularly for those organic liquids with a low surface tension. Moreover, these solid surfaces are only one time use. Taken together, the acoustic levitation method can hold the liquid droplet in air to realize lossless analyte enrichment in any liquids, which is much better than any existing solid surfaces.

Comment 1: *The idea of using acoustically levitated droplets for sample preparation in analytical chemistry has been proposed already by E. Welter et al back in 1997: E. Welter, B. Neidhart “Acoustically levitated droplets — a new tool for micro and trace*

analysis" *Fresenius J Anal Chem* (1997) 357:345-350

The abstract reads as follows: "First experiences with the technique of acoustical levitation of droplets in the field of analytical and atmospheric chemistry are reported. Acoustical levitation enables the contactless handling of solid and liquid microsamples. This avoids adsorption of analyte at and desorption of contaminants from container walls especially for liquid samples. Common experiments of sample preparation procedures were conducted in levitated drops like liquid/liquid extractions, solvent exchange, and analyte enrichment by evaporation of the solvent. A first approach was made to use acoustical levitation for the simulation of atmospheric chemistry situations."

Response 1: Thank you for your recommendation. The paper you mentioned was the first report using the acoustic levitation technique in the field of analytical and atmospheric chemistry field. We have cited this important paper in our revised manuscript. The authors used the levitating droplet aiming at prohibiting the influence of the containers. They also studied the evaporation process of a levitating water droplet, and further demonstrated enrichment of n-hexanol in methanol. After the drop was evaporated approximately half of its volume, more mixture solution of n-hexanol and methanol was introduced into the levitating drop and repeated the above processes several times. Then added a mixture of n-pentanol and methanol. Finally, the drop was transferred to a gas chromatograph to determine the content of n-hexanol in the mixture solution. These contents are the "analyte enrichment by the evaporation of the drop" mentioned in the abstract of the paper.

It is different from our analyte enrichment aiming at concentrating chemicals of extremely low concentration (*e.g.*, biomolecules, pesticide, pollutants, etc.) from the levitating drop by more than 20000 times for SERS detections. We did a lot of studies to enable the volume of sample to reduce from 10 μ l to \sim 0.5 nl. After the enrichment using the acoustic levitation technique, the detection limit reached 7.05 aM (at least four orders of magnitude lower than the SERS technique without using the droplet levitation enrichment). Moreover, the coffee ring effect could be completely solved after introducing the acoustic levitation technique into the SERS sensors. We also

demonstrated multiphase detection using the levitating droplet composed of water and organic liquids. Multiphase SERS detection is challenging (see *Nat. Mater.* 2013, 12, 165-171), which can be easily achieved after using the acoustic levitating droplet. Moreover, we showed *in situ* SERS analysis of solid objects (*e.g.*, sesame seed slice and sand powder) by applying Au nanoparticle colloidal solutions onto the levitating solid objects. After ethanol evaporated, SERS analyses of the solid objects could be directed performed. We further demonstrated concentrating and SERS detection of airborne molecules using a levitating droplet composed of Au nanoparticles to absorb the molecules in air. Our studies are important to push forward the development of the acoustic levitation-SERS sensing field pioneered by Prof. Lendl.

Comment 2: *In-situ synthesis of SERS active nanoparticles in levitated droplets and their use for trace analysis has been shown already in 2003. That paper is clearly superior to the concept presented by X. Chen as it shows a way to perform all required analytical sample manipulation, including SERS synthesis and addition of the analyte, in an automated fashion:*

*N. Leopold, M. Haberkorn, T. Laurell, J. Nilsson, J.R. Baena and J. Frank and B. Lendl
“On-line monitoring of airborne chemistry in levitated nanodroplets: In-situ synthesis and application of SERS-active Ag-sols for trace analysis by Raman spectrometry”
Anal. Chem. 75 (2003) 2166-2171*

The abstract reads as follows: “We report a new strategy for on-line monitoring of chemical reactions in ultrasonically levitated, nanoliter-sized droplets by Raman spectroscopy. A flow-through microdispenser connected to an automated flow injection system was used to dose picoliter droplets into the node of an ultrasonic trap. Taking advantage of the flow-through characteristics of the microdispenser and the versatility of the automated flow system, a well-defined sequence of reagents could be injected via the microdispenser into the levitated droplet placed in the focus of the collection optics of the Fourier transform Raman spectrometer. In that way, chemical reactions could be carried out and monitored on-line. The developed system was used for fast, reproducible, in situ synthesis of a highly active surface enhanced Raman scattering

(SERS) sol resulting from the reduction of silver nitrate with hydroxylamine hydrochloride in basic conditions. With this chemical system, SERS substrate preparation could be achieved at room temperature and in short time. The in situ prepared silver sol was used for trace analysis of several organic test molecules that were injected into the levitated SERS-active droplet again using the microdispenser. The concentration dependence of the SERS spectra was studied using 9-aminoacridine, revealing that down to the femtogram region high-quality SERS spectra could be obtained. Additionally, SERS spectra of 6-mercaptopurine, thiamine, and acridine were recorded in the levitated drop as well.”

Response 2: Thank you for your valuable suggestions. We have appreciated this Prof. Lendl’s pioneering work on the combination of SERS and acoustic levitation in our revised manuscript. This paper developed a unique automated sample loading microdispenser system to the acoustic levitator, enabling different chemical reactions to take place and to be monitored on-line by the SERS technique. Silver sol solutions were prepared in the levitating drop and used for analyses of several organic test molecules. The lowest detection concentration of 9-aminoacridine, 6-mercaptopurine, thiamine, and acridine were 2×10^{-10} M, 2×10^{-6} M, 4×10^{-4} M and 4×10^{-5} M, respectively.

Different from the 2003 paper and other previous studies, the novelty of our work lies in analyte enrichment within a levitating droplet by evaporating the solvent for ultrasensitive SERS detection. The concentration could be increased by more than 20000 times using a 10 μ l levitating droplet, which is different to achieve using other methods. We also demonstrated multiphase detection using the levitating droplet composed of water and organic liquids. Multiphase SERS detection is challenging (see *Nat. Mater.* 2013, 12, 165-171), which can be easily achieved after using the acoustic levitating droplet. Moreover, we showed *in situ* SERS analysis of solid objects (*e.g.*, sesame seed slice and sand powder) by applying Au nanoparticle colloidal solutions onto the levitating solid objects. After ethanol evaporated, SERS analyses of the solid objects could be directed performed. We further demonstrated concentrating and SERS detection of airborne molecules using a levitating droplet composed of Au nanoparticles

to absorb the molecules in air. The above contents have never been reported before. The analyte enrichment using an acoustically levitated droplet for SERS detections will attract broad research interest from the acoustic levitator and SERS community.

Comment 3: *With respect to the “coffee ring” effect X. Chen only cites his own work leading to the conclusion that the coffee ring effect can not be avoided. However, the group of Gerwert in Bochum, Germany had proposed a solution to this problem already in 2013:*

Julian Ollesch, Steffen L. Drees, H. Michael Heise, Thomas Behrens, Thomas Bruening and Klaus Gerwert “FTIR spectroscopy of biofluids revisited: an automated approach to spectral biomarker identification” Analyst, 2013, 138, 4092

In their paper they write: “A compact, benchtop sized robotic spotting system (Instrument2, M2 Automation GmbH, Berlin, Germany) was used to dispense the samples in quadruplicate on 384-well silicon multi-well titerplate (MTP) substrates (Bruker Optics, Ettlingen, Germany). Substrates were cleaned with sodium hypochlorite solution and plasma treatment (Zepto, Diener plasma surface technology, Ebhausen, Germany) before use. Each well covers a circular area of 4 mm diameter, which was almost perfectly covered with the sample. A total volume of 3 ml was loaded at a syringe-pump controlled speed into the dispenser. The single sample lm was formed from approximately 50 nl. Subsequently, the sample loaded substrate plates were vacuum-dried for 10 min.”

They continue later on: “Due to the extremely low volumes spotted, the formation of a coffee ring as still observed with nanoliter volumes was avoided. An atomic force microscopic (AFM) scan confirmed that at such a low volume, an indented cylinder shape is formed instead of an irregular ring structure (Fig. 5A and B).”

Response 3: Thank you for your comments. The paper you mentioned applied a robotic dispensing system with a piezo-driven capillary dispenser head for distributing biofluid samples on the MTP silicon substrate. To avoid produce spectral artefacts and obtain the reproducible homogeneous dry films of biofluids, a 4 mm well consisted of 217

drops of approximately 200 μl each. Then stated “Due to the extremely low volumes spotted, the formation of a coffee ring as still observed with annoliter volumes was avoided”. This paper reduced the volume of the applied liquid to achieve even distribution of analytes. To increase the SERS sensitivity, we would like to enrich the analytes from a large amount of highly diluted solutions (*e.g.*, 10 μl) into a small dot (ideally 100 μm in diameter considering this is the smallest size that naked eyes can observe and it is only slightly larger than the laser spot size of sever micrometers) with evenly distributed analytes. The analyte concentration increased thousands of times from the droplet to the small dot on the slippery surfaces. *Therefore, the method introduced in this paper cannot be used in analyte enrichment for improving the SERS detection limit.*

The acoustic levitation platform can losslessly enrich analytes from a large volume of highly diluted solutions (*e.g.*, 10 μl) composed of any volatile solvents. The concentration can be increase > 20000 times, significantly improving the SERS sensitivity. We advanced the development of the acoustic levitation-SERS sensing system, and demonstrated its ideal analyte enrichment performance for ultrasensitive, multiplex, and multiphase SERS detections.

Reviewer comments, second round –

Reviewer #1 (Remarks to the Author):

The authors have responded sufficiently to the reviewers and have now correctly referenced the original work on SERS within levitated droplets, which puts their work in the proper context. ie. That the DLE system is the new development that adds sensitivity and flexibility to the original in situ work. They have also provided more information on the limitations of the technique and enough information for those that may follow up, reproduce and use the technique.

Reviewer #2 (Remarks to the Author):

The authors have addressed my comments with considered responses and importantly provided a limit of detection of the SERS acoustic levitation approach. The authors have also extensively addressed the issue of exploding droplets and explained in detail how the heating elements were used to evaporate the droplets. All in all I am satisfied with these responses.